# Recent Advances in the Enzymatic Synthesis of Polyester

**DOI:** 10.3390/polym14235059

**Published:** 2022-11-22

**Authors:** Hong Wang, Hongpeng Li, Chee Keong Lee, Noreen Suliani Mat Nanyan, Guan Seng Tay

**Affiliations:** 1Bioresource Technology Division, School of Industrial Technology, Universiti Sains Malaysia, Penang USM 11800, Malaysia; 2Tangshan Jinlihai Biodiesel Co. Ltd., Tangshan 063000, China; 3Bioprocess Technology Division, School of Industrial Technology, Universiti Sains Malaysia, Penang USM 11800, Malaysia; 4Renewable Biomass Transformation Cluster, School of Industrial Technology, Universiti Sains Malaysia, Penang USM 11800, Malaysia; 5Green Biopolymer, Coatings & Packaging Cluster, School of Industrial Technology, Universiti Sains Malaysia, Penang USM 11800, Malaysia

**Keywords:** polyester, enzyme, enzyme-catalyzed, polymerization, solvent-free, synthesis, lipase

## Abstract

Polyester is a kind of polymer composed of ester bond-linked polybasic acids and polyol. This type of polymer has a wide range of applications in various industries, such as automotive, furniture, coatings, packaging, and biomedical. The traditional process of synthesizing polyester mainly uses metal catalyst polymerization under high-temperature. This condition may have problems with metal residue and undesired side reactions. As an alternative, enzyme-catalyzed polymerization is evolving rapidly due to the metal-free residue, satisfactory biocompatibility, and mild reaction conditions. This article presented the reaction modes of enzyme-catalyzed ring-opening polymerization and enzyme-catalyzed polycondensation and their combinations, respectively. In addition, the article also summarized how lipase-catalyzed the polymerization of polyester, which includes (i) the distinctive features of lipase, (ii) the lipase-catalyzed polymerization and its mechanism, and (iii) the lipase stability under organic solvent and high-temperature conditions. In addition, this article also focused on the advantages and disadvantages of enzyme-catalyzed polyester synthesis under different solvent systems, including organic solvent systems, solvent-free systems, and green solvent systems. The challenges of enzyme optimization and process equipment innovation for further industrialization of enzyme-catalyzed polyester synthesis were also discussed in this article.

## 1. Introduction

Polyester is a common polymer compound characterized by the existence of ester linkages in the chain’s primary structure, obtained by the gradual polycondensation of hydroxyl-containing compounds and carboxyl-containing compounds [1,2], which ranks fourth in nature after polysaccharides, proteins, and DNA. In recent years, a series of new polymeric materials have been developed rapidly due to their non-toxic and stable characteristics, for instance, polyester. Polyester is mainly divided into two categories of thermoplastic saturated polyester and thermosetting unsaturated polyester. Traditional saturated polyesters, which include polybutylene terephthalate (PBT) and polyethylene terephthalate (PET) that have high mechanical strength, abrasion resistance, corrosion resistance, and temperature resistance, as well as other outstanding properties, have been widely used as synthetic fibers and engineering plastics. With the “white pollution” intensified, polylactic acid (PLA), polybutylene succinate (PBS), and other bioplastics have been designed and manufactured [3,4]. These bioplastics have good biodegradability, biocompatibility, and non-toxicity characteristic, and they can be an alternative material to replace traditional plastics in medical and healthcare industry [5,6]. Meanwhile, thermosetting unsaturated polyester consists of an unsaturated group in addition to ester linkages in the main polymer chain, which is responsible for crosslinking process when a free radical initiator and crosslinker are added. Thermosetting polyester is an insoluble polymer with high rigidity, resilience, corrosion resistance, and weatherability, which has a wide range of applications in various industries as coatings and adhesive materials. At present, both types of polyester are mainly synthesized using the chemical polymerization method.

## 2. Current Polyester Synthesis Method and a Possible Alternative

The process of producing polyester via the chemical synthesis route requires a high temperature (above 180 °C) which may lead to excessive oxidation or even significantly damage the stability of product performance. Such high reaction temperatures would produce undesirable side reactions when using monomers with low thermal or chemical stability. As a result, this polymerization method restricts the types of polymers that can be formed [7]. Conventional methods for synthesizing aliphatic polyesters usually use alkoxy aluminum, zinc-like, and tin-like metal-containing oxides or organic compounds as catalysts [8,9,10]. This toxic metal catalyst normally might negatively affect the biosafety and biocompatibility of the end product if it cannot be removed after the process of making the polymer. Additionally, a significant quantity of organic solvents in the production process may cause environmental pollution (solution polymerization) if the management of the organic solvent is inappropriate. Generally, the hydrocarbon solvent is used to manufacture polyester, such as xylene or benzene, to avoid the viscosity buildup during the polymerization and to remove water produced from the esterification process via azeotropic distillation.

Polyesters can also be synthesized by enzyme catalysis. The Nobel Prize Winner Sumner proved in 1947 that enzymes are proteins with catalytic properties [11]. Some studies also discovered that enzymes have the advantages of high catalytic efficiency, mild reaction conditions, nontoxicity, and reusability [7,12]. In recent years, owing to the development of functional organic polymer materials and the apparent advantages of enzyme catalysts, enzyme-catalyzed synthesis of organic macromolecules has been actively researched, where enzyme-catalyzed polymerization is an essential route for polyester production [13].

There are many studies have been previously conducted to produce polyester using an enzymatic process due to several advantages, such as (i) enzymes are derivable from renewable resources; (ii) mild reaction conditions where thermal degradation of the raw materials or product can be avoided; (iii) no metal catalyst residue which makes the product achieve better biocompatibility; (iv) stereoselective, regioselective, and chemoselective, usually only a single substrate or a class of structurally similar substrates can participate in the reaction, retaining the active function of other groups of the monomer, avoiding unwanted side reactions [14]; and (v) biodegradable polyester can be synthesized using enzymes as a catalyst. As a comparison with chemical methods, enzyme-catalyzed polymerization of polyesters is a green process, which may be considered an alternative route for polyester synthesis. This process is seen as promising to resolve some issues caused by polyester synthesis, such as the disposal of unwanted side reactions as well as the metal catalyst residues. Additionally, the process may have added-value if the synthesis involves renewable resources and materials.

However, enzyme-catalyzed synthesis may have disadvantages such as high cost, large enzyme consumption, and a lesser number of high molecular weight polymers that can be produced [15]. With the development of the enzyme-catalyzed synthesis of polyesters, the challenges are believed to be addressed when more attempts are conducted. The current research on enzyme-catalyzed polymerization of polyesters mainly focuses on the types of enzymes and polymerization methods. This article focuses on the main routes of enzyme-catalyzed polyester synthesis and the increasingly attractive lipase catalysts used for polymerization, summarizes the progress of enzyme-catalyzed polyester synthesis in different solvent systems, and discusses the development trend to provide a comprehensive reference for further research.

## 3. Enzyme-Catalyzed Synthesis of Polyesters

There are several types of reaction involved in the enzyme-catalyzed polyester mechanism, including enzyme-catalyzed ring-opening polymerization (eROP), enzyme-catalyzed polycondensation, and their combination.

### 3.1. Enzyme-Catalyzed Ring-Opening Polymerization

The eROP has been widely studied since the new type was first reported in 1993 [16,17]; some simple eROP are shown schematically in Figure 1. Cyclic esters (lactones) can be easily polymerized via eROP, and no alcohol or water is produced during the reaction. Thus, no by-product removal techniques need to be considered, and a high molecular weight of the product could be yielded [18,19,20,21]. Lars van der Mee et al. [22] achieved the polymerization of lactones with 6- to 13- and the 16-membered ring using Novozym 435 as catalyze via eROP. The number average molecular weight (Mn) of the synthesized polyesters ranged from 6600 g/mol to 23,600 g/mol. It was also found that Novozym 435 has a similar affinity for all lactones, and the polymerization rate of catalytic macrocyclic lactones is greater than that of small cyclic lactones. In addition, the eROP of large ring-size lactones can be catalyzed by lipases, which is difficult to achieve by metal catalysis methods [23]. Manzini et al. [24] successfully polymerized more than a dozen giant macrolides (minimum 12-membered ring, maximum 84-membered ring) via *Candida antarctica* lipase B (CALB)-catalyzed eROP at 70 °C, and products with higher molecular weights can be obtained with longer polymerization times. Furthermore, eROP allows for the copolymerization of two or more different cyclic monomers. Srivastava et al. [25] synthesized a high molecular weight copolymer of 1,5-dioxepan-2-one (DXO) and ε-caprolactone (CL) through Lipase CA-catalyzed eROP at 60 °C. Scanning electron microscopy (SEM) analysis showed that the scaffolds of the copolymer were highly porous and contained interconnected pores. The porous structure of the samples considerably altered the tensile and viscoelastic characteristics of the scaffolds which are free from toxic metallic residue. As known, polylactones are widely applied in the pharmaceutical, biomedical, and packaging sectors because of their excellent biodegradability and biocompatibility [26,27,28]. With the growing demand for environmentally friendly, safe, non-toxic, and subsequently, functionalized polylactones, the eROP of lactones is gaining attraction as an alternative method to traditional synthesis methods [29,30,31,32].

### 3.2. Enzyme-Catalyzed Polycondensation

In the early 1980s, the first enzyme-catalyzed polycondensation was reported [33,34], and ever since, lipases have become recognized as significant tools for polymer material production. Enzyme-catalyzed polycondensation plays a crucial role in polymerization as the monomers could have a wide range of selectivity, and the products could also have diversity [7], as schematized in Figure 2. Hydroxy acids, as simple monomers containing one hydroxyl group and one carboxyl group, can be polymerized to polyhydroxyalkanoates (PHAs) by enzyme-catalyzed polycondensation. Ohara et al. [35] studied the enzymatic polycondensation of lactic acid using Novozym 435, and the high-yield oligomeric PLA was obtained. The enzymatic polycondensation of diols and diacids is commonly studied, for example, Mahapatro A et al. [36] used CALB to catalyze the polycondensation of adipic acid and octanediol for 24 h at 70 °C, and the Mn of the product could reach 17,800 g/mol. Meanwhile, Mahapatro et al. [37] compared the enzyme-catalyzed polycondensation of various diols and various diacids in a study. The results showed that the long-chain sebacic acid with octanedioic acid could be reacted faster as compared to others. Meanwhile, adipic acid and octanedioic acid could produce a polymer with the highest molecular weight among them. Daniel et al. [38] synthesized linear polyesters of hexadecanedioic acid and octanediol using the Novozym 435 enzymatic polycondensation method to produce polyester films, which were found to have multivacant surfaces, exhibiting potential value in medical or pharmaceutical fields. The enzymatic polycondensation reaction also produces water or alcohols, and the lower boiling point alcohols are easy to be removed from the reaction system than water, thus driving the equilibrium toward catalysis, and a higher molecular weight product could be obtained [39]. Therefore, in order to make the enzymatic polycondensation reaction easier, diacid derivatives are often used as monomers [40,41,42,43]. Azim et al. [44] utilized lipase to catalyze the polycondensation of diethyl succinate and butanediol to produce PBS with a weight average molecular weight (Mw) of 38,000 g/mol after a two-step reaction at temperatures of 80 °C and 95 °C, respectively. Jiang et al. [45] utilized diethyl succinate and dimethyl itaconate as the diacid derivative monomers with 1,4-butanediol to synthesize poly(butylene succinate-co-itaconate) (PBSI) via Novozym 435 catalyzed enzymatic polycondensation under vacuum condition to eliminate the alcohol generated by the reaction. PBSI with an Mn reaching 13,288 g/mol with 90% yield was obtained, and the unsaturated group was well preserved. On the other hand, hyperbranched polyol polyester could be produced from polyol using the enzyme-catalyzed polycondensation method [46,47,48]. Hyperbranched polyol polyesters are elastomers mainly used to replace soft tissues, and glycerol polyesters are among the most studied polyesters [49,50]. Kulshrestha et al. [51] studied the enzyme-catalyzed polycondensation of glycerol, adipic acid, and 1,8-octanediol at different durations, ranging from 5 min to 42 h. The results indicated that a significant interaction between monomers was observed after 45 min. The results indicated that the growth of branched glycerol polyesters occurs after linear growth. Taresco et al. [52] studied the Novozym 435 catalyzed polycondensation of glycerol and divinyl adipate to produce polyglycerol adipate (PGA) at different temperatures. The results showed that the degree of branching of PGA increased from 5% to 30% when the temperature was increased from 50 °C to 70 °C. According to Giovanni B et al. [53] the degree of branching of poly (glycerol sebacate) (PGS) can be adjusted to 56% when the substrate ratio of sebacic acid/glycerol was increased to 1.5/1.0 in the CALB catalyzed enzymatic polycondensation reaction. Therefore, the degree of branching of PGS can be controlled by adjusting the substrate ratio to meet the needs of medical applications. Enzymatic polycondensation offers a greener way to synthesize polyesters from many bio-based monomers, which could reduce environmental and health impacts, and therefore, enzymatic polycondensation has received considerable attention in recent years [3,4,39,54].

### 3.3. Enzyme-Catalyzed Copolymerization

Enzyme-catalyzed copolymerization can be employed to prepare multi-block copolymers (Figure 3), which can change the properties of copolyesters in a wide range to obtain polymer materials for different applications. Kobayashi et al. [55] had a study to polymerize carboxylic anhydride monomer and ethylene glycol via enzyme-catalyzed copolymerization, and a high yield of polymer was obtained. In later studies, carboxylic anhydride monomers were expanded to include anhydrides of various diacids [56,57]. Jiang et al. [58] successfully used Novozym 435 to catalyze the enzyme-catalyzed copolymerization of ω-pentadecalactone (PDL), diethyl succinate, and 1,4-butanediol to synthesize a new ternary polyester. The Mn of the copolymer could be reached 24,400 g/mol, and the polymer was stable up to 300 °C with less than 0.1% weight loss.

## 4. Overview of the Biological Catalyst: Lipase

Enzymes are natural catalysts with a wide variety of thousands of species, and it has been widely used in studying various enzyme-catalyzed reactions. Enzymes can be used to decompose macromolecules and also to prepare macromolecular materials for various applications [59]. Lipases, amylases, and proteases are hydrolytic enzymes. They can catalyze various reverse hydrolysis reactions under nonaqueous conditions and prompt the formation of a substantial number of chemical bonds to form polymers, such as polyesters, polysaccharides, and polypeptides, and these are some examples of the macromolecules produced using an enzyme as a catalyst [60]. Lipases (triacylglyceride hydrolases EC3.1.1.3) are the third most commercially accessible enzymes after carbohydrases and proteases [61]. They currently represent more than twenty percent of the worldwide market. Lipases based on animals, plants, and microbes are essential industrial enzymes, and they are easily available. Among them, microbial lipases are widely used as industrial biocatalysts due to their abundance, rapid reproduction, wide range of temperature and pH adaptation, and ability to be isolated and purified in the form of extracellular enzymes. *Aspergillus niger*, *Bacillus subtilis*, *Candida antarctica*, *Candida rugosa*, *Pseudomonas cepacia*, *Rhizopus chinensis*, *Rhizomucor miehei*, *Thermomyces lanuginosus*, and *Yarrowia lipolytica* are the most used microorganism for lipase synthesis. CALB is one of the most popular and developed lipases due to its broad substrate scope, high polymerization catalytic activity, high thermal stability up to 100 °C (immobilized form), and its stereoselectivity for small and large secondary alcohols [62,63,64,65,66].

### 4.1. Lipase Catalyzed Hydrolysis: Structure and Mechanism

The catalytic properties and capacity of lipases are inextricably linked to the structure of lipases. Most lipases regulate the activation and suppression of catalytic activity at the interface between lipids and water by opening and closing the removable sequence motifs [67]. The movable sequence motifs are called “lids,” which are essential for the catalytic activity of lipases and protect the active group’s structure [68]. It was observed that when the lipase hydrophilic side of the cap faces outward in a pure aqueous medium, the lid is closed to protect the catalytic unit. In contrast, in an aqueous medium containing oil, the lid is open and turned over with the lipophilic side facing outwards, exposing the catalytic unit, and showing catalytic activity. Khan F.I. et al. [69] analyzed the lid structures of 149 different lipases in the Protein Data Bank and classified them into different categories based on their structures, as depicted in Figure 4, such as no lid, one circular or helix lid, and two or more helixes lid. It is also found that lipases in the same category have similar substrate specificity and mechanism of action, e.g., monoglyceride and diglyceride lipases have one small loop lid [70], triacylglycerides lipases have one lid [71], thermophilic lipases have two or more helical large lids [72]. In other words, the substrate preference, stability, and other properties of lipases can be predicted based on the lid’s structure. Catalytic properties, thermal stability, and reaction efficiency of lipases can be enhanced by modifying the lid on the targeted mutation and lid swapping, thus guiding the large-scale industrial application of lipases.

Lipases are a large class of enzymes that can hydrolyze esters at the oil-water interface with a high catalytic rate. When the lipase lid is opened, the catalytically active triad is exposed to interact with the substrate to complete the catalytic reaction. As shown in Figure 5, amino acid fragments consisting of Ser-Asp-His form the catalytic triad, and the proton transfer chain formed between them contributes to the hydrolysis reaction. First, histidine absorbs a proton from serine, and serine binds the carbonyl oxygen atom to form an enzyme-substrate tetrahedron [77,78]. This tetrahedron is characterized by the oxygen atom at the center of the ester bond. The tetrahedron forms an acyl-enzyme intermediate and an alcohol group in the presence of water before the histidine passes the absorbed proton to the acyl-enzyme intermediate. The enzyme releases the substrate and regenerates it. The next ester is then catalyzed by the free enzyme, and the same activity is repeated, which the process is called the “ping-pong mechanism” [79]. The hydrolysis principle of lipase is widely used in the hydrolysis and synthesis of esters and amide groups.

### 4.2. Mechanism of the Lipase-Catalyzed Polyester

The synthesis of polyester using lipase as a catalyst follows the mechanism of the lipase hydrolysis reaction. The hydrolysis reaction is reversible subject to the environment, as lipases can catalyze esterification or transesterification reactions under organic solvents or low aqueous conditions. Depending on the different monomers used, lipase-catalyzed polyester synthesis reactions include two major categories, respectively, eROP and enzyme-catalyzed polycondensation [81].

The catalytic reaction mechanism of eROP(lipase) is shown in Figure 6. As depicted, serine combines with cyclic ester monomers to form an enzyme-substrate active intermediate, (ii) enzyme-substrate active intermediate is unstable and rapidly hydrolyzed under the action of water or alcohols to achieve both ring-opening of substrate and recovery of lipase, (iii) after which, the recovered enzyme (lipase) combines with other monomers to form another enzyme-substrate active intermediate, the ring-opening molecule will replace water to act on it while forming dimer and free enzyme, (iv) then, the dimer continues to work on the new enzyme substrate-active intermediate to obtain the triploid, and finally achieve ring-opening polymerization [82].

As shown in Figure 7, the enzyme-catalyzed polycondensation reaction involves the condensation of monomers with multiple carboxyl groups and hydroxyl groups to form polymers, which are catalyzed by an enzyme. As step-growth polycondensation proceeds, small molecules, such as water and alcohol, are liberated through gradual ligation reactions between substrates to form new chemical bonds. The enzyme-catalyzed polycondensation mechanism is similar to the enzyme-catalyzed ring-opening polymerization. In the active site, the enzyme binds the monomer (polycarboxylic acid or ester) to form an enzyme-substrate complex and release by-products (water is produced if the monomer is polycarboxylic acid, or alcohol is produced when the monomer is an ester formed from polycarboxylic acid and alcohol). The complex reacts with another monomer consisting of a hydroxyl group (polyol) to form a dimer and releases an enzyme. The dimer has the ability to combine with enzymes to form enzyme-substrate complexes; or to react directly with enzyme-substrate complexes to become oligomers, which may polymerize to produce a polymer [82]. As the enzyme links the monomers together, hydrolysis (reverse reaction) may be taken place due to the increase of by-product amount. However, the growth of the polymer may be terminated when the equilibrium has been achieved, even though the monomers are not completely consumed. Thus, the yield of the enzyme-catalyzed polycondensation and the molecular weight of the product can be increased by removing the by-product from the system, and the reverse reaction can also be avoided [83].

### 4.3. Stability of Lipase in the Synthesis Reaction

Lipases are frequently used as biocatalysts in organic synthesis attributed to the benefits of enzymatic reactions carried out in organic solvent systems as opposed to the aqueous ones, such as providing a reaction medium for substrates or products with low water solubility and reducing undesirable hydrolysis side reactions. Therefore, it is important to understand the stability and activity of lipase in organic solvents. In each solvent, the lipase activity behaves differently. Water molecules are bound in and around the enzyme molecule by hydrogen bonds, which play an essential role in protein folding, structure, and activity [84]. The studies conducted by Zaks et al. [85,86] showed that the interaction of organic solvents with enzyme-bound water affected the enzyme activity and that the stability and activity of the enzyme were higher in non-polar solvents than in polar solvents. This phenomenon was also observed in lipase [87,88,89]. Zahid Kamal et al. [90] studied the effects of six polar solvents on 6B lipase and found that the lipase maintained its structural integrity after the solvent volume concentration was increased to 60%, and the solvent did not penetrate into the interior but bound to the active site of the enzyme, reducing the catalytic capacity of the active site for the substrate, and leading to enzyme inactivation. Peter Trodler et al. [91] investigated the behavior of CALB in polar and non-polar solvents, respectively. The results showed that polar solvents tend to interact directly with bound water on the enzyme surface, while non-polar solvents enhance the interaction between neighboring water molecules on the enzyme surface, leading to a restricted surface water exchange. In non-polar organic solvents, the electrostatic effect of lipase is significantly enhanced, and the electrostatic properties are very close to those under vacuum conditions [92]. The lid of the active structure of lipase opens in the presence of a non-polar solvent, giving lipase its catalytic capacity [93]. As a comparison of the catalytic reaction rates of lipase in polar and non-polar organic solvents, it was found that the latter was greater than the former [94]. Enzyme technology under an organic solvent system is currently used in pharmaceuticals, food modification, biodiesel, and other industries [95,96,97,98,99]. In order to obtain higher commercial value, the search for highly tolerant lipases with organic solvents has never stopped. Priyanka et al. [100] screened a solvent-tolerant *Listeria monocytogenes* that secretes a solvent-tolerant lipase that loses only 20% of its activity after 24 h in a 30% volume of methanol. Yasir Ali et al. [101] isolated a new strain of *Pseudomonas* spp. in an alkaline environment. The lipase produced by the new species showed significant resistance to dimethyl sulfoxide (DMSO), ethanol and tolerated various solvents at low-temperature conditions. Solvent-tolerant lipases greatly improve the application limitations of lipases, and facilitate the existing applications as well as new ones.

Another critical feature of lipases is their temperature stability. High temperatures cause proteins’ hydrogen bonds and hydrophobic interactions to break down, which causes protein denaturation and aggregation [102,103]. In addition, chemical changes are also one of the causes of high-temperature inactivation. K. Bhanuramanand et al. [104] incubated lipase 6B at 75 °C and found that there was no change in protein structure, but rather a deamidation of asparagine resulted in the inactivation of the lipase. The lipase can withstand a small temperature range, and the enzymatic reaction is often limited by temperature conditions [105]. In a general polymerization process, high-temperature conditions could facilitate the solubility of the substrate and shorten the reaction time while lowering reaction media viscosity. Therefore, improving the thermal stability of lipases is favorable for industrial applications, and heat-resistant lipases are one of the desirable targets for research on extreme enzymes [106,107,108]. Kamal et al. [109] screened a lipase 6B from *Bacillus subtilis*, and it was noticed that the melting temperature (Tm, temperature at which 50% of the protein is unfolded) is 78 °C, and the optimum reaction temperature is 65 °C, respectively. Castilla et al. [110] isolated *Janibacter* sp. strain R02 from Antarctic soil. The extracellular lipase of this strain has been identified with enzyme activity at 80 °C, and it has high thermal stability at 100 °C. According to them, there was almost no loss of enzyme activity after it was incubated at 100 °C for 1 h. *Pseudomonas* spp. strains have been found to produce an extracellular super heat-stable lipase with activity at 90 °C and an average half-life of about 13 h at this temperature [111]. Meanwhile, with the development of genetic engineering, studies on modifying lipases by gene fusion and targeted mutation to achieve thermal stability improvement of lipases have become highly debated nowadays. Cai Haiying et al. [112] performed codon optimization modification of *Thermomyces lanuginosus* lipase (TLL) and recombinant expression in Pichia pastoris. The optimal reaction temperature of the lipase was determined to be 60 °C; Zhang et al. [113] recombinantly expressed the heat-resistant lipase of *Streptomyces thermophilus* in *Escherichia coli*, and the optimal temperature of the reaction was 60 °C, which could be considered to have a good thermal stability. Fang et al. [114] had successfully cloned and expressed lipase from *Thermomicrobium roseum* DSM 5159 in *Escherichia coli* and *Bacillus subtilis*, which has a good thermal stability. Owing to their thermostability and applicability at higher temperature ranges, it is believed that high-temperature active lipases are excellent biocatalysts in the production of polyesters.

## 5. Solvent System for Enzyme-Catalyzed Synthesis of Polyesters

The enzyme-catalyzed synthesis of polyesters can occur by solution polymerization or bulk polymerization. The yield, molecular weight, and subsequent application of the products are related to the selection of the polymerization reaction system. Solvent systems for the enzyme-catalyzed synthesis of polyesters are usually divided into organic solvent systems and solvent-free systems. In most studies, solvents are used to bring the reactants together in a uniform phase to enhance the rate of those chemical reactions. In solvent-free systems, the enzyme is directly dispersed in the reagent to catalyze the polymerization reaction [115]. In addition, “green” media systems such as water, supercritical carbon dioxide, and ionic liquids are those attractive solvents for reactions involving biocatalysts and have been employed in the enzymatic process for polyester synthesis [116].

### 5.1. Organic Solvent System

In organic solvent systems, the substrate and the enzyme are dissolved in a solvent, which makes the substrate more accessible to and from the active site, thus improving the efficiency of enzymatic polymerization reactions [117]. The solubility of the substrate and polymer needs to be considered while choosing the right organic solvent in the synthesis, as it can improve the solubility of the substrate as well as reduce the solubility of the polymer product. Additionally, the removal of the enzyme after the reaction can be facilitated by the solvent [96]. Generally, organic solvents can influence the catalytic activity of lipase. For instance, dimethylformamide and dimethyl sulfoxide can dissolve and inactivate the lipase. Hence, the selection of the organic solvent must be appropriate according to the needs [118]. In a solvent-based system, factors such as the solvent’s solubility in water, dielectric constant, polarity, and log P (P = octanol to water partition coefficient) should always be considered since they may promote or inhibit the polymerization process. Generally, the log P, which is a qualitative parameter, can be used to determine the impact of solvents on catalytic activity. Solvents with low log *p* values are hydrophilic that exhibit enzyme inhibition, whereas solvents with high log *p* values are hydrophobic solvents, which favor highly active enzymes in nonpolar compounds [119]. Despite the fact that polar solvents contribute to the solubility of some monomers, their hydrophilic character may lead to the loss of the hydrated layer of the enzyme, which can result in its inactivation or denaturation, hence reducing its catalytic activity [120]. Therefore, solvents with low polarity are often selected for enzymatic synthesis (high log *p* value, between 1.9 and 4.5). The commonly used solvent in the enzymatic process and its log *p* values are shown in Table 1. Furthermore, for reactive groups to be accessible, the 3D shape of the binding site must have a high enzyme-solvent affinity [121]. The physicochemical properties of solvents have to be considered simultaneously throughout the reaction process, especially the boiling point and chemical inertness, which avoid the consequences of solvent reduction due to low boiling point and solvent denaturation due to solvent participation in the reaction [64].

Gross et al. [64] studied the eROP of ε-caprolactone using different solvents. It was found that toluene was beneficial to the stability of Novozym 435, and the highest molecular weight of the polymer can be obtained at 70 °C. However, the highest polymerization rate was taken place at 90 °C with 90% conversion of the monomer, which was achieved in 2 h. Yao et al. [129] investigated the effect of different hydrophilic solvents on the Novozym 435 catalyzed polycondensation of octane diol, adipic acid, and L-malic acid. In hydrophilic solvents (acetone, tetrahydrofuran, and t-BuOH). The polymers produced using this technique have considerably lower molecular weights than the polymers produced in hydrophobic solvents (isooctane, n-hexane, and toluene). Among those hydrophobic solvents, the highest molecular weight products were obtained at 80 °C with isooctane as solvent. From a comparison study conducted by Juais et al. [130] on the Novozym 435 catalyzed synthesis of polyisosorbide adipate, it was found that the solvent accelerates ethanol removal via azeotropic distillation, resulting in a high molecular weight polymer, and the highest molecular weight was obtained in a solvent mixture of cyclohexane: benzene (6:1 *v/v*). Linko et al. [131] studied the effect of high boiling point solvents on polymers under conditions where lipase was used as a catalyst. The solvents used were xylene, diphenyl ether, dodecane, 2-methoxyethyl ether, triethylene glycol dimethyl ether, tetraethylene glycol dimethyl ether, hexyl ether, isoamyl ether, and o-phenylene dimethyl ether. It was found that diphenyl ether was the most suitable solvent for this study, and it produced a polymer with the highest molecular weight. Additionally, there have been many attempts to use solvent systems in an enzymatic process to conduct polymerization, and the detail is depicted in Table 2.

### 5.2. Solvent-Free System

In a solvent-free environment, the enzyme is wholly disseminated throughout the reagents. It catalyzes the native polymerization of the monomer [115], which can act directly on the reaction substrate. Hence, it can be said that the solvent-free system has the advantages of [145,146,147]: (i) the high concentration of reaction substrate; (ii) fast reaction speed; (iii) small reaction volume; and (iv) reduced steps for product separation (no solvent recovery process). The lack of organic solvents drastically reduces environmental contamination and reduces the cost of organic solvent recovery. Therefore, enzyme-catalyzed synthesis of polyesters in solvent-free systems is a promising clean reaction technology. However, the lack of solvent causes poor mass transfer of the reactants and a dense reaction system, which restricts the even growth of the polymer [44,148].

Juais et al. [130] utilized Novozym 435 as a catalyst for the synthesis of polyisosorbide adipate in a solvent-free system. The condition employed produced low molecular weight polyesters with an Mn value of 974 g/mol at a reaction temperature of 85 °C for 168 h. Kato et al. [149] produced a polymer made of 1,6-dihydroxyhexane and dimethyl 2-mercaptosuccinate under a solvent-free condition using Novozym 435 as a catalyst. From the results, polyesters with the Mw of 14,000 g/mol were produced at 70 °C after 48 h. Ivone et al. [150] had a study to utilize Novozym 435 to catalyze polycondensation of seven dibasic acids and six diols in a solvent-free environment. The result has shown that long carbon chain monomers such as sebacic acid and 1,8-octanediol were easier to be polymerized than short-chain monomers. The polycondensation reaction of sebacic acid and 1,6-hexanediol yielded a high molecular weight polymer with Mw of 27,121 g/mol after 96 h at a temperature of 90 °C. Zhao X et al. [151] studied a solvent-free condition using lipase to catalyze diethyl glutarate, ethylene glycol diacetate, and cotton samples; it was found that synthetic poly(ethylene glutarate) was linked to the molecule of the cotton surface at 40 °C after 9 h, resulting in the in situ coating on the cotton. It has increased the hydrophobicity of the cotton, while other physical properties were not affected much. The study also showed that the solvent-free system on an enzyme-catalyzed synthesis of polyester is a simple and environmentally friendly method. Filippo Fabbri et al. [40] conducted a study on the enzymatic catalyzed polycondensation of dimethyl itaconate and 1,8-octanediol in a solvent-free system. A polymer with an Mn of 4100 g/mol was obtained when the reaction temperature was set at 85 °C for 24 h. Nguyen et al. [152] successfully used enzymes to catalyze the polycondensation of glycerol, azelaic acid, and tall oil fatty acid to synthesize highly branched polyesters with Mw up to 39,700 g/mol, and the polyesters the films made of this polyester have high hydrophobicity with a glass transition temperature of around −33 °C. Kening Lang et al. [46] synthesized poly(glycerol-1,8octanediol-sebacate) (PGOS) from sebacic acid, glycerol, and 1,8-octanediol using Novozym 435 as a catalyst under solvent-free conditions. The hyperbranched polymer POGS, with relative Mn and Mw of 9500 and 92,000 g/mol, has shown outstanding electrospinnability. Solvent-free enzymatic polymerization without metal and solvent residues is a potential synthesis method for biomaterials. There were attempts to use a solvent-free system on polymerization via an enzymatic process as shown in Table 3.

### 5.3. Green Solvent System

In addition to solvent and solvent-free systems, water, supercritical carbon dioxide (ScCO_2_), and ionic liquids (ILs) can also be used and facilitated enzyme-catalyzed polymerization reactions. Water is the prototypical green solvent, being non-toxic, non-flammable, and cheap. The use of water as a solvent in enzymatic polymerization has economic and environmental benefits as compared to organic solvent media. Kobayashi et al. [167] proved that both eROP and enzymatic polycondensation can be carried out in an aqueous system. The enzymatic polycondensation process of sebacic acid and 1,8-octanediol in water at 45 °C for 24 h had synthesized a polymer with the Mn of 1600 g/mol, whereas the eROP of lactones in water at 60 °C has formed polymers with the Mn ranged from 500 to 1300 g/mol after 72 h. Namekawa et al. [168] found that the enzymatic polymerization of macrolide was fast in an aqueous system and could form stable emulsions with water, while caprolactone did not show any reaction in the system. Pfluck A.C.D et al. [169] developed a synthesis method of poly(octamethylene octanoate) (POS) in an aqueous system via enzymatic polycondensation. The Mw of POS had reached 6900 g/mol at 45 °C after 48 h, and the enzyme activity was stable in water. It was noticed that there was no significant change after repeating the usage for seven cycles. This study has indicated that enzymatic polymerization in an aqueous media system has potential for industrial applications.

ScCO_2_ is a benign fluid above its critical point. Due to its inert, non-toxic, non-flammable, and recyclable properties, which is believed could be a potential replacement for organic solvents [170,171,172]. Polloni A.E.d et al. [173] produced polyester with a molecular weight of 33,000 by enzyme-catalyzed ROP of ω-pentadecalactone in ScCO_2_ after 2 h of reaction, and the water content of the reaction medium was found to be inversely proportional to the molecular weight. Domenico et al. [174] used ScCO_2_ as the solvent and Novozym 435 as the biocatalyst for the low-temperature polycondensation of cis-9,10-epoxy-18-hydroxyoctadecanoic acid (CHA) at 35–55 °C to obtain polyesters with an average molecular weight of 18,000. The ScCO_2_ solvent can be an alternative way to ease the mass transfer between reactants under low-temperature conditions, and retain the valuable groups of monomers during the polymerization process. It can be wholly separated from the final product by pressure relief at the end of the reaction without the greenhouse effect, and can be used repeatedly by compression without corrosiveness [175]. Owing to the abovementioned, it can be said that ScCO_2_ is a promising new technology for various chemical reactions, but the development of pressurization devices and the safety of reaction equipment must be in consideration before it can be widely used to produce polymer in an industrial scale.

ILs, as known as molten salts, have low vapor pressure, strong chemical stability, and attractive solvation properties for many organic or inorganic chemicals, which have gained widespread recognition as novel solvents in enzymatic polymerization [26,176]. The enzyme can maintain its excellent thermal stability and activity in the ILs system [26,177]. Piotrowska et al. [178] conducted the eROP of rac-lactide and ε-caprolactone in ILs (1-butyl-3-methyl-imidazolium tetrafluoroborate, [BMIM][BF4]) using CALB. A copolymer with an Mn of 3100 g/mol was obtained after 14 days at a reaction temperature of 80 °C. Curie et al. [179] measured the effects of six organic solvents and ILs (1-butyl-3-methyl-imidazolium hexafluorophosphate, [BMIM][PF6]) on eROP. The results have shown that the highest molecular weight and conversion were obtained in the ionic liquid [BMIM][PF6], which were 3108 g/mol of Mn and 93.2%, respectively. ILs can also be configured as “ tailored solvents “ for different reaction requirements because their properties depend on their anion-cation combinations, and can be fine-tuned [180]. Zhao et al. [181] synthesized a series of glycol-functionalized ILs to obtain low-viscosity salts, which were validated by Novozym 435 catalyzed eROP of L-lactide. It was shown that the ILs yielded poly(L-lactide) with a higher Mw of 23,000 g/mol than the solvent-free system (Mw = 12,400 g/mol). This study has indicated that ILs can be used as a substitute for volatile organic solvents in eROP reactions, and they can be used repeatedly. It can also be modified by cationic or anionic units to adjust the melting point and viscosity of ILs, making them an ideal solvent for enzymatic polymerization.

## 6. The Challenges of Enzymatic Synthesis of Polyester

In recent years, enzymatic polymerization has received considerable attention in making polymer products for various applications due to its advantages. However, there is still room for further study on the feasibility of this process, such as the process efficiency, parameters to be determined for process optimization, reactor enhancement, and cost of the enzyme. As a comparison to traditional batch enzymatic polymerization reaction, the reaction in flow could often obtain high molecular weight and conversion in a short period. For instance, Wissal Adhami et al. [142] had synthesized poly(δ-valerolactone) (PVL), poly (ε-caprolactone) (PCL), and PVL/PCL block copolymer by eROP in flow. As a result, the yield of PCL reached 100% within a period of 4 min at 70 °C, whereas the yield of PVL reached 100% within a period of 10 min at room temperature, and the yield of synthetic copolymer reached 93% after 4 min at 70 °C. It was also found that the catalyst (Novozym 435) could be reused at least 10 times with retention of its high catalytic performance. On the other hand, WanYou et al. [182] prepared PCL at different reaction temperatures and reaction times using CALB. An Adaptive Fuzzy Inference Systems (ANFIS) model was developed using 42 samples obtained from the experiment, and the model could predict the PCL molecular weight well. In a solvent-free system, the rate of enzymatic polymerization might be slow because of the mass transfer limitation. However, ultrasound may be able to overcome this drawback. Zhao et al. [183] synthesized poly (ethylene glutarate) via immobilized CALB-catalyzed polycondensation in a solvent-free system; the reaction time was reduced from 24 h to 7 h when ultrasound treatment was employed. In the synthesis of polyester via enzymatic process, there are factors that need to be considered to avoid catalyst deactivation, the high cost, stability under temperature and polarity conditions, suitability for reaction substrates, reaction products, and enzyme reusability. Idris et al. [184] reviewed various immobilization methods to stabilize the catalytic activity of CALB and pointed out that activated polystyrene nanoparticles were economical and practical materials for the immobilization of CALB. Montanier et al. [185] redesigned the active site of CALB and applied it to the synthesis of ε-caprolactone, which resulted in a three-fold increase in catalytic efficiency and 1.3 times Mn of polyester could be produced as compared to CALB.

## 7. Conclusions

Enzyme-catalyzed polymerization, including through eROP, enzyme-catalyzed polycondensation, and enzyme-catalyzed copolymerization, is a powerful tool for the synthesis of green polyester materials. Enzymatic synthesis of bioplastics such as PLA and PBS, which could avoid high temperature and metal oxide employment, is a more environmentally friendly pathway. Meanwhile, enzyme-catalyzed polymerization can avoid chemically unstable monomers and thermally unstable monomers side reactions, such as the retention of double bonds in unsaturated polyester, which enable the resin to have further interaction, i.e., cross-linking process. Concomitantly, the chemoselectivity of enzymes allows functional groups in polymerization without cumbersome protection/deprotection steps, resulting in the development of diverse functional polyesters. In addition, an enzymatic method is also a powerful tool for synthesizing a modified polyester. Enzyme-catalyzed polymerization can also accomplish the task of modification and grafting biomolecule substrates to obtain a new polymer. Enzymatic synthesis of polyesters with a wide range of substrates and products often requires the design of an overall strategy based on conditions and purpose. This article has systematically reviewed the recent advances in lipase-catalyzed polyester synthesis, including the lipase and the polymerization mechanism; the different polymerization types; the advantages and disadvantages exhibited in different media (organic solvents, solvent-free, aqueous solutions, supercritical fluids, and ionic liquids); and explores the key issues for large-scale mass production. Non-organic solvent for enzyme-catalyzed polymerization has been developing rapidly, and more research is needed on this promising method to optimize the parameters. It is believed that enzyme-catalyzed polymerization will be the key to polymer synthesis in the future.

## Figures and Tables

**Figure 1 polymers-14-05059-f001:**
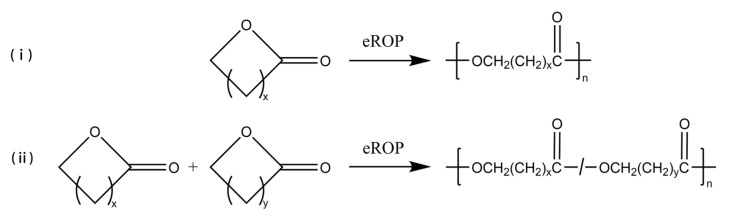
The eROP of (**i**) lactones, (**ii**) different lactones. x and y represent the number of carbon atoms in straight-chain or branched-chain, x, y > 0.

**Figure 2 polymers-14-05059-f002:**
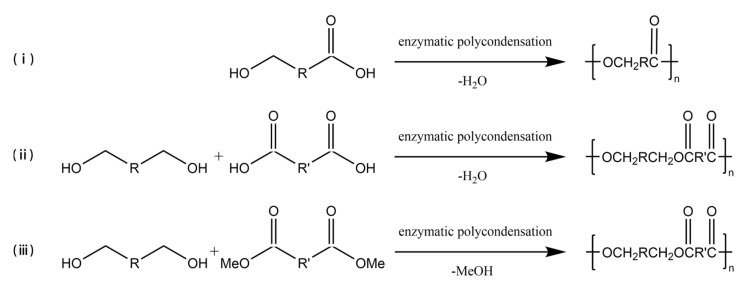
The enzyme-catalyzed polycondensation of (**i**) hydroxyacids, (**ii**) diols and diacids, and (**iii**) diols and diacid derivatives.

**Figure 3 polymers-14-05059-f003:**
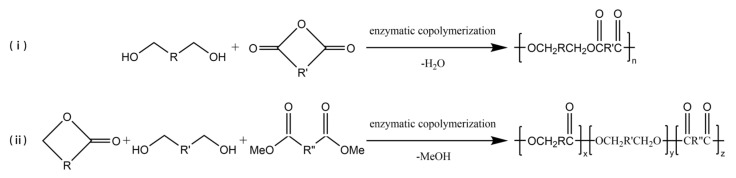
The enzyme-catalyzed copolymerization of (**i**) diols and carboxylic anhydride (**ii**) lactones, diols, and diacid derivatives.

**Figure 4 polymers-14-05059-f004:**
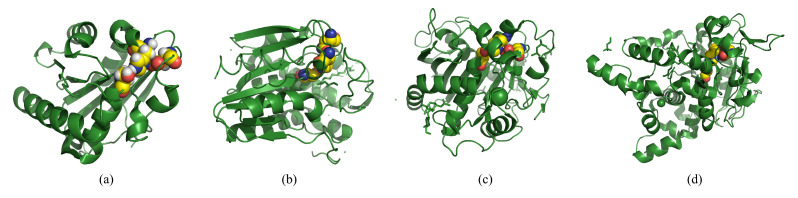
Lipases with different lid structures. (**a**) Crystal structure of lipase A from Bacillus subtilis (PDB entries 5CT5) without lid domain, adapted with permission from Rdf. [73]. Copyright 2015, John Wiley and Sons (**b**) Crystal structure of Monoglyceride lipase from Bacillus sp. H257 (PDB entries 4KE6) with a loop lid domain, adapted with permission from Rdf. [74]. Copyright 2013, Elsevier (**c**) Crystal structure of Lip2 lipase from Yarrowia lipolytica (PDB entries 3O0D) with a helix lid domain, adapted with permission from [75]. Copyright 2010, Elsevier (**d**) Crystal structure of Lipase from Geobacillus thermocatenulatus (PDB entries 2W22) with two helices in its lid domain, adapted with permission from [76] Copyright 2009, Elsevier. The red, yellow, and blue spheres in the diagram are the active catalytic residues of lipase. The copyright detail is shown in Appendix A.

**Figure 5 polymers-14-05059-f005:**
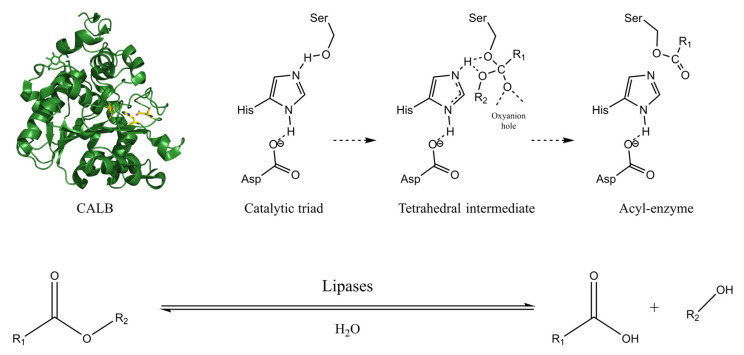
Mechanism of the hydrolysis reaction of ester bonds catalyzed by lipases. The Asp-His-Ser catalytic triad of CALB (PDB entries 4ZV7) [80] is represented by the red, yellow, and blue ball-and-sticks. The copyright detail is shown in Appendix A.

**Figure 6 polymers-14-05059-f006:**
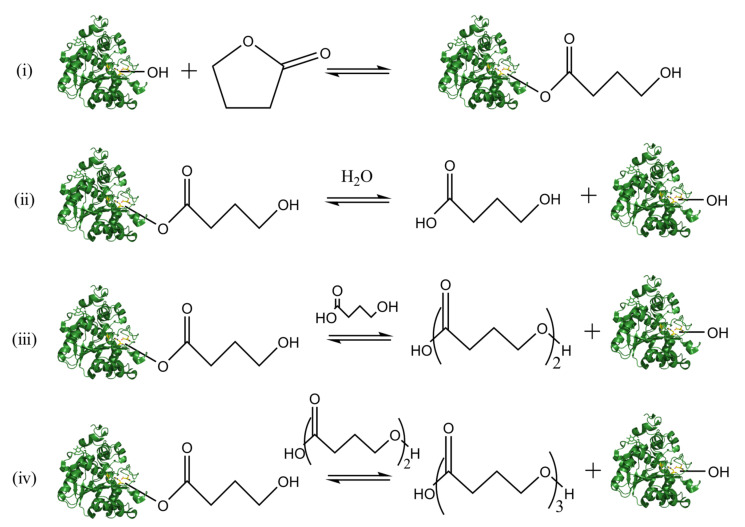
Lipase-catalyzed ring-opening polymerization (in the case of butyrolactone, butyrolactone is polymerized by eROP). (**i**) the formation of enzyme-substrate active intermediate, (**i**) the hydrolysis of enzyme-substrate active intermediate, (**ii**) the formation of dimer, (**iv**) the formation of triploid. Structures of lipases were rendered using PyMOL [80]. The copyright detail is shown in Appendix A.

**Figure 7 polymers-14-05059-f007:**
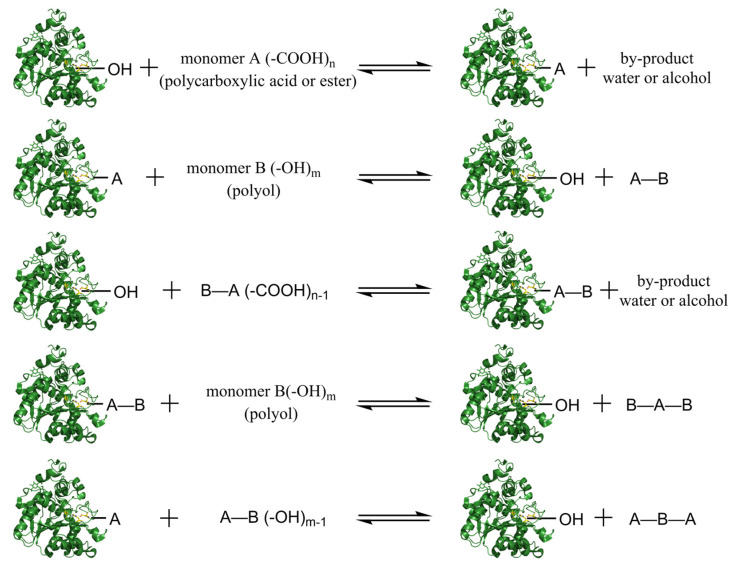
Enzyme-catalyzed polycondensation in the presence of Lipase. monomer A represents polycarboxylic acid or ester formed with alcohol, “n” represents the number of carboxylic acid groups in monomer A. monomer B represents polyol, “m” represents the number of hydroxyl groups in monomer B (m ≥ 2, n ≥ 2). Structures of lipases were rendered using PyMOL [80]. The copyright detail is shown in Appendix A.

**Table 1 polymers-14-05059-t001:** Log P of some organic solvents.

Solvent	Log P	Boiling Point (°C)	Reference
diphenyl ether	4.21	258	[122]
p-cymene	4.1	177	[123]
n-heptane	4	98.4	[124]
cyclohexane	3.71	80.74	[122]
n-hexane	3.5	68.73	[122]
toluene	2.5	110.6	[125]
2-MeTHF	1.85	80.2	[126]
t-butyl methyl ether	1.43	55.2	[124]
tetrahydrofuran	0.53	66	[127]
Cyrene	−1.52	226	[123]
dimethyl sulfoxide	−1.35	189	[124]
ethyl ether	−0.83	34	[124]
acetone	−0.21	56	[128]

**Table 2 polymers-14-05059-t002:** Enzyme-catalyzed synthesis of polyesters in solvent.

Enzyme	Monomer	Temperature (°C)	Solvent	Mw (g/mol)	Mn (g/mol)	Ref
immobilized *Humicola insolens*	brassylic acid, 1,8-octanediol, ω-hydroxyhexadecanoic acid	70	diphenyl ether	-	40,400	[132]
Novozym 435	unsaturated/epoxidized, diacid (C18, C22), diols	90	diphenyl ether	57,000	-	[133]
Novozym 435	dimethyl adipate, 1,4-butanediol, 1,8-octanediol	85	tetrahydrofuran	2200	-	[134]
immobilized CALB	diester, aromatic diol	85	diphenyl ether	2000– 4000	-	[135]
immobilized CALB	azelaic acid, 1,8-octanediol	75	toluene	27,000	21,000	[119]
Novozym 435	L-malic acid, 1,3-propanediol, 1,5-pentanediol, or 1,8-octanediol	40	dimethylformamide	4800– 8100	-	[136]
Novozym 435	adipic acid, 1,8-octanediol	70	diphenyl ether	-	28,500	[37]
*Candida antarctica* Lipase	polynonadic anhydride, 1,8-octanediol	60	toluene	-	10,000	[137]
Novozym 435	adipic acid divinyl ester, glycerol	50	tetrahydrofuran	-	5200–13,000	[52]
Novozym 435	adipic acid, L-malic acid, 1,8-octanediol	80	isooctane	17,400	-	[129]
Novozym 435	dimethyl itaconate, 1,4-butanediol, diethyl adipate	80	diphenyl ether	-	94,000	[138]
Novozym 435	dimethyl itaconate, butanedioic acid, butanediol	80	diphenyl ether	-	1948–13,288	[45]
Novozym 435	diethyl furan-2,5-dicarboxylate dodecane-1,12-diol, diethyl oxalate	100	diphenyl ether	-	8900	[139]
CALB	diethyl succinate, 1,4-butanediol dilinoleic diol	95	diphenyl ether	-	16,300–25,200	[140]
immobilized CALB	suberic acid, 1,8-octanediol	60– 80	cyclohexane: tetrahydrofuran 5:1 *v/v*	16,000–19,800	-	[141]
CALB	ferulic-based diester, 1,4-butanediol	90	diphenyl ether	-	2000	[42]
Novozym 435	δ-valerolactone	85–95– 110	toluene	-	2300	[142]
Novozym 435	trans, β-dimethyl hydromuconate, adipic acid, 1,8-octanediol	85–95– 110	diphenyl ether	-	21,900	[143]
CALB	cis-tetraphenylporphyrin macrocycle	70	toluene	-	9600	[144]

**Table 3 polymers-14-05059-t003:** Enzyme-catalyzed synthesis of polyesters in no-solvent.

Enzyme	Monomer	Temperature (°C)	Solvent	Mw (g/mol)	Mn (g/mol)	Ref
Novozym 435	unsaturated/epoxidized, diacid (C18, C22), diols	90	no-solvent	25,000	-	[133]
Novozym 435	1,12-dodecanedioic, 1-thioglycerol	80	no-solvent	5380	-	[153]
Novozym 435	1,18-cis-9-octadecenedioic, glycerol	90	no-solvent	-	6000–9100	[154]
Novozym 435	dicarboxylic acid diesters and diols	85	no-solvent	-	1094- 11,549	[155]
immobilized CALB	1,4-butanediol, dimethyl adipate	70	no-solvent	2200	1000	[156]
lipase from *Candida sp.99-125*	diethyl sebacate, 1,4-butanediol	70	no-solvent	-	15,800	[157]
Novozym 435	tall oil fatty acid, glycerol, azelaic acid	90	no-solvent	20,900–39,700	-	[152]
Novozym 435	dimethyl itaconate, 1,4-butanediol, dimethyl succinate	60	no-solvent	-	840	[158]
Novozym 435	itaconic acid, diethylamine, 1,8-octanediol	85	no-solvent	-	696	[159]
Novozym 435	dimethyl itaconate, 1,4-cyclohexanedimethanol	50	no-solvent	-	720–2859	[160]
Novozym 435	itaconic acid, succinic acid, sorbitol	90	no-solvent	-	1140–11,900	[161]
Novozym 435	ε-caprolactone	90	no-solvent	-	3100	[162]
Novozym 435	dimethyl succinateand, 1,4-butanediol, ε-caprolactone.	90–110	no-solvent	-	7100–10,100	[163]
CALB	divinyladipate, 1,4-butanediol	70	no-solvent	31,000	7100	[164]
CALB	1,4-butanediol, diethyl succinate	40	no-solvent	-	2800	[165]
Novozym 435	1,6-hexandiol and diethyl adipate	100	no-solvent	-	5000–12,000	[166]

## Data Availability

Not applicable.

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
