# Peer review of "Recent Advances in the Enzymatic Synthesis of Polyester"

_polymers, 2022, doi:10.3390/polym14235059_

Round 1
Reviewer 1 Report
Review report:
Manuscript ID: polymers-2026166
The aim of the review article on “Recent advances in the enzymatic synthesis of polyester” by Hong Wang, Hong Peng Li, Lee Chee Keong, Noreen Suliani Mat Nanyan and Tay Guan Seng to present the reaction modes of enzyme-catalyzed ring-opening polymerization, enzyme-catalyzed polycondensation, and their combinations, respectively. The challenges of enzyme optimization and process equipment innovation for further industrialization of enzyme-catalyzed polyester synthesis were also discussed in this article. The authors carried out a well-planned review with
interesting results. The set of data presented is enough and the literature was critically explored to support the authors' hypotheses. The paper falls
within the scope of the journal of Polymers. Obtained results are novel and surely deserve to be published in the journal after minor revision.
My comments are detailed below:
The abstract should be more informative.
The English should be improved in both the correct uses of terms and syntax.
Some grammar and punctuation in the text:
Line 16. - please add “the” before “automotive industry”.
Line 19. – add “an” before “organic solvent”.
Line 19. - delete “,” at the end of the sentence.
Lines 20 – Please write “methods”.
Line 21. – delete “as” and write “a new ….”
Line 23. – delete “on” at the end of line 23.
Line 24 – Please write “lipase catalyzed” as “lipase-catalyzed”
Line 28. – please write all 3 “system” as “systems”.
Line 54. – please add “the” before “chemical”.
Line 57. – please add “the” before “chemical synthesis”.
Lines 62-63. – please write “aluminium” as “aluminum”.
Line 80. – delete “which”.
Line 91. – delete “as”.
Line 92. – add “as” before “promising”.
Line 95. – add “and” before “materials”.
Line 97. – please add “a” before “lesser”.
Line 104. - please add “a” before “comprehensive”.
Line 105. - please add “a” before “comprehensive”.
Line 115. - please add “the” before “product”.
Line 118. – delete “were” before “ranged”
Line 133. – Change “its” to “their”.
Lines 134-135. – Change “subsequent” to “subsequently”.
Line 171. – Change “a” to “an” before “Mn”.
Line 191. - Change “consirable” to “considerable”
Line 353. – write “low temperature” as “low-temperature”.
Line 391. - please add “the” before “polymerization”.
Line 464, 481, and line 536. – Change “a” to “an” before “Mn”.
Line 488. – change “condition” to “conditions”
Line 492. – change “enzymatic” to “an enzymatic”
Line 551, 591. - please delete “the” before “enzymatic”.
Line 551. - please add “the” before “enzyme”.
Line 576. - please add “the” before “immobilization CALB”.
Line 591. - please delete “the” and add “an” before “enzymatic”.
Line 593. – write “substrate” as “substrates”.
Line 593. - please add “a” before “new polymer”.
Line 602. – please change “for” to “to” before “polymer synthesis”
In General – in all sentences thought text before the “and” please put “,”.

Author Response
Dear Reviewer,
Thank you very much for the comment, and I appreciate it very much. Kindly find attached the responses to the comment.
Thank you

Reviewer 2 Report
The Authors’ Review is devoted to the actual and interesting field of obtaining polyesters using "green chemistry", i.e. through the use of enzymes as catalytic systems.
The manuscript is well written, contains a sufficient amount of reference data and examples. Thus, the manuscript makes it possible to learn about some of the latest advances in the enzymatic synthesis of polyesters.
However, there are a number of remarks.
1) Authors often argue that traditional polyester synthesis (via metal catalysts) usually requires the use of large amounts of organic solvents. This is not entirely true: the industrial synthesis of the most common polyesters proceeds in the melt and, further, under the conditions of solid-state polycondensation. At the same time, the authors point out that the enzymatic synthesis of polyesters proceeds most efficiently in a medium of non-polar organic solvents. This doublethink makes the argument that enzymatic synthesis is more environmentally friendly less valid. I believe that the wording in the text related to the above phrases needs to be corrected.
2) Authors often use the term "polymerization" when it comes to unambiguous "polycondensation", i.e. interaction of hydroxy- and carboxy- groups with the release of a low molecular weight by-product. For example, on line 35 (but not only there!). The term polymerization can be used, for example, in the case of the formation of PLA from lactide, or in the opening of other lactone rings.
3) In fig. 6 and 7 there are highlights of the text with a yellow marker.
4) On line 404 there is a text highlight in blue.
5) In table 1, the names of solvents and images of their structural formulas are very poorly readable: you need to increase their size and / or clarity, or leave only the name, without formulas, because these solvents are well known.
6) On line 502, the designation of degrees Celsius is incorrect.
In general, I believe that this manuscript can be published with minor changes.
Author Response

(The authors gave the same response as above.)
